# Improving the Efficiency of Non-Stationary Climate Control in Buildings with a Non-Constant Stay of People by Using Porous Materials

**DOI:** 10.3390/ma14092307

**Published:** 2021-04-29

**Authors:** Alexander Shkarovskiy, Shirali Mamedov

**Affiliations:** 1Construction Network and Systems Department of Koszalin University of Technology, 75-453 Koszalin, Poland; 2Department of Heat and Gas Supply and Ventilation, Saint Petersburg State University of Architecture and Civil Engineering, 190005 St. Petersburg, Russia; 3Department of Metal and Timber Constructions, Saint Petersburg State University of Architecture and Civil Engineering, 190005 St. Petersburg, Russia; mamedov_am@bk.ru

**Keywords:** porous materials, channel porosity, internal climate, non-constant stay of people, heating, energy conservation

## Abstract

This article presents the results of experimental research on the non-stationary management of the internal climate of buildings with a non-constant stay of people. During the absence of people, a significant drop in air temperature and corresponding energy conservation in heating is possible. The effectiveness of porous building materials is shown, provided that the appropriate characteristics are selected. Daily fluctuations in the outside temperature are completely extinguished by a layer of foam polystyrene insulation. The absence of channel porosity in the structural material of the wall is a guarantee of the stability of its thermal and humidity regime. This, in turn, prevents the development of mold and mildew.

## 1. Introduction

The use of porous materials can provide new and unexpected possibilities in problems that are considered to be already solved [1]. However, an important factor is the use of a material with correctly selected characteristics [2]. Our article presents the results of using such materials for non-stationary climate control (NSCC) in non-residential buildings. The term NSCC refers to the programmed control of heating, ventilation, and air conditioning systems in non-residential buildings during the absence of people. In this case, the temperature in the rooms is not kept constant, but changes depending on the presence of people.

Heating costs are as significant for the individual consumer as on a national scale. Residential and public buildings account for 41% of total energy consumption, of which almost 60% are the costs of heating and ventilation [3]. Thus, energy conservation in heating is a natural tendency wherever it does not lead to a decrease in the quality of life.

The most inefficient use of energy is observed in buildings with a non-constant stay of people. This category includes mostly public buildings—schools, universities, office buildings, shopping malls, etc. In such places, it is natural to control the indoor climate actively [4]. During the absence of people, there is no need to comply with the thermal comfort requirements; consequently, a significant decrease in air temperature is possible. This is highly efficient and it is a very important, inexpensive, and easily implemented way to increase the energy efficiency of buildings [5]. In the case of two-stage regulation (Figure 1), before the end of the working day, it is enough to turn off the heating system so that, in some cases, up to 5% of the system capacity is left to prevent pipelines freezing.

A controlled decrease in air temperature begins in the premises. It is limited only by technical requirements (safety of equipment, plants, works of art in museums, etc.) [6]. If there are no special requirements, then it is important to prevent moisture condensation on the enclosing structures. From this point of view, the minimum permissible temperature is $t_{int.}^{\min}$ = 10 °C. Before the arrival of people (workers, students, visitors, etc.), the heating is switched on in advance. Provided that it is technically possible, the heating system is switched on with increased power. This will ensure that the temperature rises faster by the start of the working day. The difference indicated by the colored areas in the figure below in the time–power coordinates is a geometric representation of daily energy conservation.

This method of energy conservation has been known for a long time [7]; however, its widespread adoption has been hindered for several reasons. Firstly, there was no theoretical basis and method for active microclimate control calculations. The authors have carried out the necessary theoretical studies in this area. An engineering method for calculating the programmed regulation of a heating system was also developed [8]. The method allows the stages of temperature reduction and subsequent heating to be calculated in relation to the size and design of a building.

Secondly, this method of energy conservation is often confused with room temperature regulation according to heating schedules [9]. The use of heating schedules is a stationary regulation method as it is aimed at long-term maintenance of a certain temperature in rooms. In fact, in this case, the heating system operates with a reduced but constant heat output, depending on the outside and inside temperatures. In the case of active climate control, the heating system is turned off completely, but for a limited time, when there are no people in the premises.

The third reason is the widespread belief that a decrease in indoor temperature can lead to an undesirable change in the temperature distribution in building envelopes. This fact, in turn, can cause subsequent saturation of the enclosing structures with moisture and, as a result, mold and fungal crops can appear.

The last reason should be recognized as the most significant one. The reason for the cases of negative experience of using the NSCC method, according to the authors, is due to a simplified approach to the use of insulation materials. By default, all porous materials are considered to be good thermal insulation materials, though they are not only characterized by high thermal resistance.

Unlike other building materials, they are characterized by a different combination of various thermophysical characteristics. Along with the thermal conductivity coefficient, these include density, thermal diffusivity, porosity, hygroscopicity, and a number of others [10,11]. It is the combination of these characteristics that is important in this case.

In this case, the influence of porosity on moisture permeability is especially important [12,13]. Porous materials have a varied structure, which largely determines the possibility of moisture penetration. There are several types of porosity such as general, closed, and open as well as channel porosity. Channel porosity is especially important from the point of view of moisture permeability since it depends on the closed porosity weakly and is considered to be an independent parameter [14]. Moreover, it is the porosity that determines the gas and moisture permeability of a material [15]. The predominance of channel porosity can be accompanied by moisture migration in building materials, which can lead to undesirable consequences [16,17]. Excess moisture on the wall can even unpredictably affect the operation of heating system devices [18,19]. In this case, the presence of moisture can indeed lead to the appearance of mold and fungal crops. Moisture in microchannels can remain for a long time, even with the next heating of the wall [20]. This may also be facilitated by the formation of hydrates in a limited space of pores [21,22].

In addition, the presence of moisture in microchannels and micropores can have unexpected effects on thermal conductivity. Recent studies have shown that the transfer of the interaction between the solid and the liquid phase contained in the pores to the capillary level and the appearance of hydrogen bonds can dramatically increase the thermal conductivity of a material [23]. It has even been proposed that the consideration of these processes be transferred to the nanoscale, when the effect of aggregation processes occurring between nanoparticles in various aqueous solutions becomes noticeable [24]. In our case, this means a noticeable decrease in the thermal insulation properties of the material and confirms the inadmissibility of the constant presence of moisture in the pores.

Another important result of the latest research is the ability to predict the properties of various materials based on machine learning methods. In this direction, we can note the successful application of the Gaussian process regression (GPR) model to predict delamination in various composites [25]. The use of such methods makes it possible to predict the relationship between predictors of properties of permeable concrete with any of the required properties, including density, porosity, and thermal conductivity [26]. This will allow porous materials with the required properties for subsequent construction to be chosen at the design stage.

In this direction, the authors have undertaken extensive experimental research. The results of one of these experiments are presented in this article. The aim of the research was to prove the safety of the NSCC method when using porous materials with correctly selected characteristics. In this case, the non-stationary control of the internal climate for a limited time, even on weekends, should not lead to a significant change in the temperature distribution in the structures. At the same time, the absence of channel porosity should prevent moisture migration.

## 2. Design and Realization of Experiments

### 2.1. Characteristics of the Research Object

For research, a typical building with a non-constant stay of people was chosen—Koszalin University of Technology (Poland). Here, people stay in rooms without outerwear. The type of work is easy mental work. The normalized air temperature in rooms is +20 °C. The city is located in a climatic zone with a design temperature for a heating design of −16 °C.

The research was carried out in several adjacent rooms on the top floor of a 7-storey building. The premises, with an area of about 16.5 m^2^, has only one external wall, facing southeast. The size of the windows (in the frame) is 1.6 × 1.6 m, and the frames are metal plastic with double glazing. The ratio of glazing area to floor area is 0.136. In each room, under the window, there is a 1072 W Purmo panel heater equipped with a thermostatic valve. The construction of walls using porous materials is very important for performing research. The four-layer structure includes a bearing layer (aerated concrete blocks), expended polystyrene insulation, and plaster on both sides, as shown in Figure 3a. The thermal and physical parameters of the design (according to the manufacturer’s specifications) are given in Table 1. A heat–humidity calculator based on the European standard was used for the calculations [27].

Thus, the heat transfer coefficient of the outer wall was ***k*** = 1/3.933 = 0.254 W/(m^2^·K). This complies with the European norms for thermal protection of buildings in force from 1 January 2014 (required value: 0.25 W/(m^2^·K)). However, this indicator is lower than the norms that came into practice on 1 January 2017 (0.23 W/(m^2^·K)) and those introduced from 1 January 2021 (0.20 W/(m^2^·K)). An important factor is the manufacturer’s guarantee of the absence of through-channel porosity of aerated concrete blocks and expanded polystyrene. In addition, the alkaline reaction of aerated concrete is an additional property that prevents the development of fungal crops.

### 2.2. Research Program

It was necessary to investigate the efficiency of porous materials in two-stage non-stationary climate control on working days (Figure 1). It was important to prove the safety of the NSCC method for enclosing structures. It was also planned to determine the optimum start time of heating so that the required temperature in the premises was reached by 8 a.m. The following experimental plan was developed when the heating system was turned off one hour before the people left (8 p.m.), and the temperature was continuously recorded from 7 p.m. to 8 a.m [28]. The measurement interval was 10 min, and the temperatures of indoor and outdoor air, temperatures on both surfaces of the wall, and at selected points inside its structure were measured.

### 2.3. Experimental Unit and Research Technique

The experimental complex was based on the AVT5330 multipoint electronic temperature recorder (Figure 2a) with software for operation in Windows. Automatic measurement is possible at any interval, starting from 2 s.

The recorder provides the connection of 8 DS18B20 temperature sensors (Figure 2b). The connection is made with a 2-m cable. For protection from external influences, the sensor is insulated with a heat-shrinkable sheath. The sensors were pre-calibrated in a certified laboratory.

In Figure 3, the layout of the sensors is shown. Sensors 2, 3, 5, and 8 were placed inside the wall in channels 8 mm in diameter at different depths. At the same time, sensor 3 recorded the temperature at the boundary of the carrier layer and the insulation (Figure 3a). Sensors 7 and 1 were fixed directly on the inner and outer surfaces of the wall, sensor 4 recorded the air temperature in the room, and sensor 6—the temperature of the outside air at a distance of about 0.5 m from sensor 1.

The sensors were placed at a sufficient distance from each other (Figure 3b) in order to avoid changes in the measurement results. Channels were drilled from inside the room, and sensors 1 and 6 were passed through the window frame. The volume of the drilled channels was negligible in comparison with the volume of the wall covered by the experiment and could not affect the heat transfer process.

Simulation of the operating modes of the heating system was carried out using Danfoss thermostatic valves. Setting the valve to the minimum position led to the shutdown of the heating device. Setting the valve to the maximum position provided heating with power of up to 150%.

## 3. Research Results and Discussion

Preliminarily, using our own methodology [29], calculations of temperature changes in the premises were performed, assuming that the outside temperature was −7 °C and there was a possibility of increasing the power of the heating system during heating up to 155%, as shown in Figure 4.

After that, experimental studies began according to the program mentioned above.

As an example, the measurement results are given for a typical day with a drop in outdoor temperature to −2.0 °C. Figure 5 shows the temperature change recorded by each sensor during the period when the heating system was turned off. Figure 6 shows the temperature profiles for the selected moments in the same period.

The temperature in the room, on the inner surface of the wall, and in its bearing layer did not undergo significant changes, despite the complete shutdown of the heating system and noticeable fluctuations in the outside temperature. Outside temperature fluctuations at night were almost completely extinguished by the layer of foam polystyrene. This is precisely the technological task of thermal insulation. An additional thermal buffer is the carrier layer made of aerated concrete blocks.

Thus, the obtained data confirm the safety of the method of active climate control for enclosing structures. The impossibility of significant penetration of a zone with a temperature below the dew point (about 6 °C) deep into the wall structure has been proved. This makes it impossible to moisten the carrier layer and prevents the further development of unfavorable phenomena (mold, fungal cultures). Thus, the effectiveness of the use of materials that do not have channel porosity was also proved.

Fluctuations in the temperature of the internal air turned out to be much fewer than the calculated ones (Figure 4). This can be explained by the fact that the calculation method considers the internal volume of the building as empty space. In reality, interior walls, ceilings, furniture, and equipment are massive heat accumulators. This softens the temperature drop after the heating system is turned off. An additional factor is the use of porous materials in the construction of the wall.

The rise in the outside temperature in the morning can be explained by the influence of solar radiation. In addition, the extreme changes in the internal temperature after turning on the heating are caused by the proximity of the heater.

Since the temperature drop in these rooms was very small, there was no need to start the heating phase ahead of time, as it was supposed by theory (see Figure 1 and Figure 2). The heating system was switched on at the beginning of the working day. The thermostatic valves were immediately set to the calculated position.

The studies were carried out every winter from 2015 to 2019, when the building structure was additionally insulated. Additional thermal protection only enhanced the achieved effect and the conclusions drawn during the research. The goal of the work was recognized as being achieved and the experiments were terminated.

## 4. Conclusions

Studies have confirmed the ease of use and high economic effect of the method of saving energy through non-stationary control of the internal climate in buildings with a non-constant occupancy of people.It has been proven that despite the complete shutdown of the heating system, there were no noticeable temperature fluctuations in the room, on the wall surface, and inside the bearing layer of the enclosing structures. Night-time fluctuations in the outdoor temperature were completely extinguished by the insulation layer, which is its technological purpose.The method does not cause moisture migration and permanent moisturizing of porous building and insulation materials. As a result, there is no danger of a significant change in the thermal insulation properties of materials and undesirable development of mold and fungal crops.An additional guarantee of the absence of adverse side effects is the use of building materials that do not have channel porosity. At the same time, modern modeling methods allow us to predict and correctly apply materials with precisely selected properties already at the design stage.The research results indicate a significantly greater effect of the NSCC method in comparison with the theoretical change in temperature. This, firstly, proves the effectiveness of the use of porous materials in the construction of walls. On the other hand, it reveals that it is necessary to make adjustments to the calculation methodology, taking into account the internal structure and equipment of the building.

## Figures and Tables

**Figure 1 materials-14-02307-f001:**
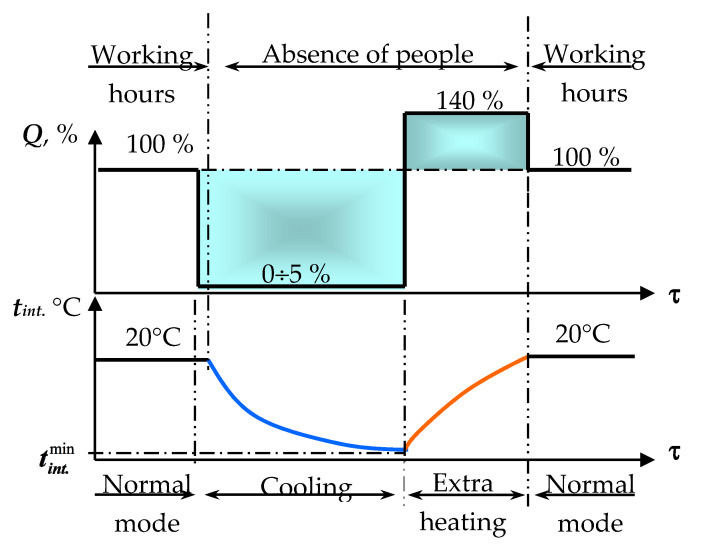
The principle of two-stage climate control (***Q***—relative power of the heating system; **τ**—time of day; ***t_int._***—internal air temperature).

**Figure 2 materials-14-02307-f002:**
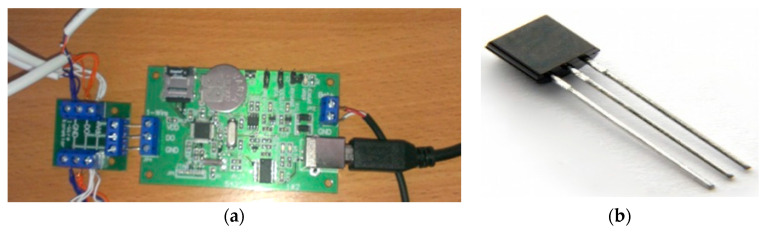
Measurement complex: (**a**) AVT5330 recorder; (**b**) DS18B20 sensor.

**Figure 3 materials-14-02307-f003:**
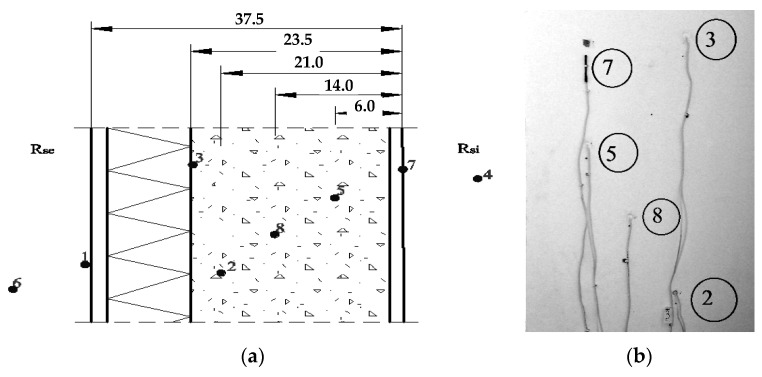
Placement of temperature sensors: (**a**) along the thickness of the enclosing structure (centimeters); (**b**) in the plane of the wall.

**Figure 4 materials-14-02307-f004:**
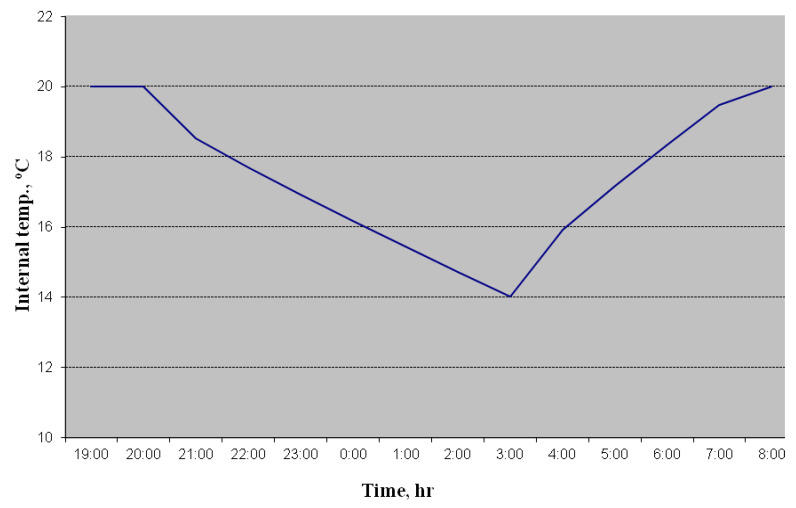
Calculated graph of non-stationary climate control on a working day at an outside temperature ***t_ext_**_._* = −7 °C.

**Figure 5 materials-14-02307-f005:**
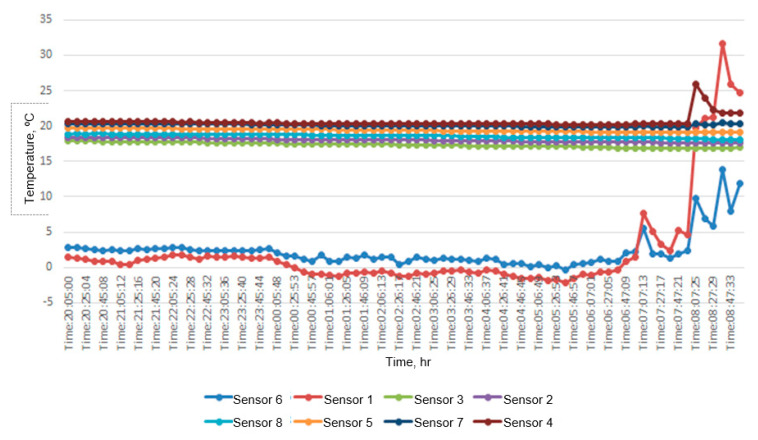
Change in sensor data for a typical measurement day.

**Figure 6 materials-14-02307-f006:**
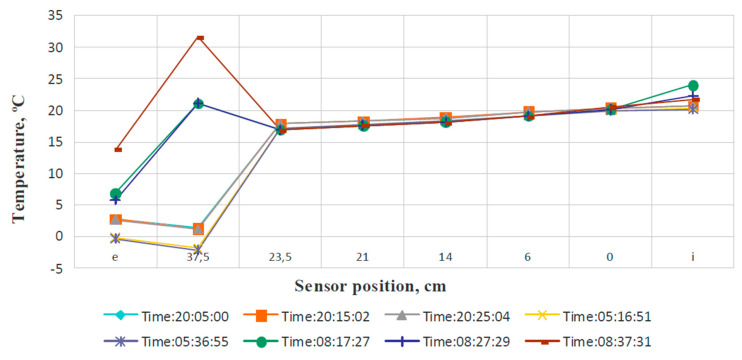
Temperature profiles for the selected measurement times.

**Table 1 materials-14-02307-t001:** Indicators of thermal protection of the external wall structure.

No. of the Layer	Layer Material or Thermal Resistance	Material Density	Layer Thickness	Coefficient of Heat Conductivity	Heat Resistance
		kg/m^3^	m	W/(m·K)	m^2^·K/W
***R_se_***	Heat exchange on the external surface	-	-	-	0.040
1	Lime-cement plaster	1850	0.015	0.82	0.018
2	Masonry of aerated concrete blocks on cement mortar	600	0.24	0.16	1.500
3	Foam polystyrene	12	0.10	0.045	2.220
4	Mineral thin-layer plaster	1480	0.02	0.8	0.025
***R_si_***	Heat exchange on the inner surface	-	-	-	0.130
	Σ	-	-	-	3.933

## Data Availability

The data presented in this study is available at the request of the respective author.

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
