# Peer review of "Improving the Efficiency of Non-Stationary Climate Control in Buildings with a Non-Constant Stay of People by Using Porous Materials"

_materials, 2021, doi:10.3390/ma14092307_

Round 1

Reviewer 1 Report

The article presents an interesting study on the non-stationary management of the internal climate in buildings with a non-constant stay of people. However, the organization and content of the manuscript need to be revised substantially before publishing. 

  1. The manuscript title needs to be changed. The title looks like a manuscript focusing on the materials synthesis, characterization and improvement, which is not the focus of the presented work. The title should change to focus on the application of the porous materials in the building, and to match the abstract and introduction. 
  2. The abstract should have less background information about benefits of porous materials used in certain situations, but rather have more presentation of the result and findings. 
  3. The introduction part has too much lengthy background on heating cost and environmental impact, which is not the focus of the work. The authors should enrich more on the materials aspects, like types of porous materials commonly used, advantages compared to non-porous ones, and current state-of-art. 
  4. The authors are recommended to cite more recent and related references, to topics like heat transfer, composite materials, civil engineering applications. The following references are suggested:
     doi: 10.1177/0021998320984245; Advances in Civil Engineering Materials 10, no. 1 (2021): 56-73; 
  5. Section 2 , 3, and 4 should be combined as one big section with subtitles. These are all related to design of experiment. The authors need to have a better organization of the work. 
  6. change subsection "research results and discussions" to simply "Results and discussions". Actually, the author should avoid using too many "research" throughout the manuscript, as every manuscript is about some research for sure. 
  7. The authors need to heavily enrich the discussion of the results part, it it too short compared to introduction and experimental setup. The authors should compare results to previous related studies, and discuss about new findings, new problems solved, and new perspectives on the topic.

Author Response

Dear Reviewer. We thank you for your deep and professional analysis of our article. Your comments and suggestions will be useful to us not only when working on this article, but also in our subsequent publication work. We are glad to inform you that all your comments and suggestions have been taken into account. Our answers are given below in accordance with the numbering of comments and suggestions in the review. Please use the adjustment and change tracking mode to see the corrections in the revised manuscript.

1. The manuscript title (lines No. 1, 2) has been changed to reflect its subject matter and better match the abstract and introduction. This change is all the more important because it coincides with the suggestion of the Reviewer #4.

2. The abstract (lines No. 9-16) has been revised to better represent the results and findings.

3&4. For the purpose of a more detailed coverage of aspects related to porous materials and the current state of the issue, the most recent publications from this area were taken into account in the introduction (new positions in the list of references are 23-26 including the two references you suggested). Thank you very much for your precise recommendation. The most significant changes in the introduction were made between lines No. 107 and 108.

5. In order to better organize the work, sections 2, 3 and 4 have been combined into one large section2. Design and realization of experiments” (before line No. 115) with subsections 2.1 (line No. 115), 2.2 (line No. 141) and 2.3 (line No. 151). The remaining sections were renumbered accordingly (lines No. 201 and 249).

6. The title of the section “Research Results and Discussions” was changed to “Results and Discussions” (line No. 201).

7. As far as possible, without significantly increasing the volume of the article, the discussion of the results and conclusions were enriched for better connection with the introduction and experimental setup. The main changes are out of the box between lines No. 256 and 259. 

Reviewer 2 Report

The manuscript is well organized and the authors have presented the research work in good manner. The readers will have great interest in the article if accepted. I suggest to accept the manuscript without any changes.

Author Response

Dear Reviewer. Thank you for your kind and professional assessment of the article. We are very pleased that this article has received such approval from a real specialist in our field of research. Your support will help our team of authors in our future work!

Reviewer 3 Report

It is good research direction to pursue. The manuscript is not appropriate to publish at current form.  

  1. The writing style of this manuscript looks like the report, rather than the paper. Needs to modify significantly.
  2. The references are not comprehensive. Suggested to add the following related to heat transfer nano fluids and materials.                                    Improved Thermal Conductivity of Fluids and Composites Using Boron Nitride (BN) Nanoparticles through Hydrogen Bonding, Thermochimica Acta, 178927, 2021

Author Response

Dear Reviewer! Thank you very much for the time you have devoted to analyzing and evaluating our work. Your comments and suggestions will be useful not only when working on this article, but also in subsequent publication work. Answers are given below in accordance with the numbering of comments and suggestions in the review. Please use the adjustment and change tracking mode to see the corrections in the revised manuscript.

  1. Sorry, but I cannot accept your comment regarding the style of the article. It's not just that it essentially requires writing a new article. Style is a relative assessment and does not indicate the level of the submitted manuscript. Style characterizes the author, is his hallmark, and not always everyone should like it. I ask you not to regard this as immodesty, but with more than 250 publications and 15 monographs and academic textbooks in my list of scientific achievements, I believe that I can afford to write in my own style. Any technical research article is a kind of research report. Therefore, I consider the style of the report one of the possible styles of the manuscript.

2. Thank you very much for the valuable recommendations on expanding the analysis and the reference you suggested! This article and several related publications were unexpected for me, but very useful! At first glance, they are devoted to the exact opposite task - to increase the thermal conductivity of materials, while we are engaged in increasing the thermal protection of buildings. However, these publications have revealed to me the additional danger of water in the pores of building and insulation materials. The transition of the interaction of substances to the capillary level and the nanoscale, indeed, can change the properties of porous materials in an unpredictable manner. My research team has already planned additional research in this area. The most recent publications from research area were taken into account in the introduction (new positions in the list of references are 23-26 including the reference you suggested). Thank you very much for your precise recommendation. The most significant changes in the introduction were made between lines No. 107 and 108.

Reviewer 4 Report

The authors present a work where the temperature profile of a wall made of porous material was monitored over 24 h cycles in order to improve the energy efficiency of the buildings. The work eventually has merit to be published in the journal materials. However, the several aspects must be improved before publication. 

1) The title is misleading because the reader assumes that several a comparison of different porous materials was made, or at least was compared with non-porous material. 

 2) Generally, the English level of the manuscript must be improved. Mainly, use conjunctions to better link the sentences. 

3) The introduction must be improved. The concepts presented in the paper should be briefly explained. For instance, the authors start stating there are many problems considered to be solved, but do not mention which ones. Generally, along the manuscript the authors require the reader to follow references to understand what they mean. The purpose of the references is to support the sentences, not to be a direct link of something that the authors want the reader to magically understand (with exceptions for technical data or specific works).  

4) On line 23: The article presents - It is not clear to which article the authors are referring to.

5) The authors should briefly explain what is NSCC.

6) on line 25, it is not defined the meaning of HVAC.

7) On line 66, the caption of the figure should be more complete. Please ad the meaning of Q. 

8) The sentence on line 108 is a bit misleading. The authors claim extensive research on the paper or generally? In this paragraph should be sharply stated what is intended to achieve in this paper. 

9) In table 1, the sum of the values in the 5th column is not 0.25. 

10) Please make a figure where it is shown the structure of the wall matching the materials in table 1. Also, were this values measured or were from some handbook? References how the values were obtained are needed. 

11) More details about the material (concrete block) lack of channels would be desirable. I understand, that would be not practical, but some kind of physical characterization of the material would be desirable. Any characterization of the porosity would be desirable (average pore size etc). 

12) The authors drill the wall to insert the sensors. How the authors avoid the hole made in the wall will not affect the temperature measurement? Any insulation was added?

13) on line 146 should be briefly explained the method, and then refer to reference 24 for additional details. The reader should not have to consult all the references in order to understand the manuscript. 

14) On line 193; caption of figure 3, it seems that sensor number 3 in figure a) is not consistent with sensor number 3 on figure b

15) On line 202:  It is cited a reference, that is not in English language, that is hard to follow and understand what the authors mean. The method should be briefly explained, if there is no English reference explaining the method.

16) figure 4 shows a graph of temperature vs time. How can this be called a designed mode?

17) In the discussion of the results, the authors should compare with similar works (if available) in the literature where channeled porous materials where studied. 

18) How these results compare with non NSCC methods? 

Author Response

Dear Reviewer. We thank you for your deep and professional analysis of our article. Your comments and suggestions will be useful to us not only when working on this article, but also in our subsequent publication work. We are glad to inform you that all your comments and suggestions have been taken into account. Our answers are given below in accordance with the numbering of comments and suggestions in the review. Please use the adjustment and change tracking mode to see the corrections in the revised manuscript.

1) The manuscript title (lines No. 1, 2) has been changed to reflect its subject matter and better match the abstract and introduction. This change is all the more important because it coincides with the suggestion of the Reviewer #1.

2) Numerous corrections have been made in the text of the manuscript using conjunctions to better link the sentences. As examples, we can cite corrections in lines Nos. 27, 28, 29-30, 30-35, 37-39, 45-46, 68-67, 72-74, 76-78, 78-79, 83-84, 88-89, 98-100, 101-102, 145-147, 176-177, 202-204 etc.

3) Thank you very much for your precise recommendations on expanding the analysis. The text of the introduction has been thoroughly analyzed and corrected (lines No. 21-113). The most recent publications from research area were taken into account in the introduction (new positions in the list of references are 23-26). The most significant changes in the introduction were added between lines No. 107 and 108.

4) You are right - the text is not specific. We have replaced "The article" with "Our article" (line number 23).

5) In accordance with your exact remark, the essence of the NSCC method was briefly explained (line No 26).

6) This is a perfectly logical remark. The abbreviation HVAC is known only to specialists in the field of heating and ventilation. Since this abbreviation is not required anywhere else in the manuscript, the full definition of heating, ventilation and air conditioning has been added to the text instead (line No 25).

7) The figure caption was supplemented with a description of the variables: Q - relative power of the heating system; τ - time of day; tint. – internal air temperature (line No. 66).

8) This sentence is really misleading. To avoid ambiguity, mention has been added to the text that only the results of only one of the experiments are given in this article: “The results of one of these experiments are presented in this article” (line No 108).

9) This is a stupid mistake, thanks for your attention. Summation was performed only in 6th column. The value 0.25 has been removed from the table (line No 131).

10) The construction of the wall, although somewhat later, is shown in Fig. 3-a. Therefore, it was decided not to give an additional figure, but only to make the appropriate link in the text (line No. 127). The values of the material characteristics given in Table 1 are taken in accordance with the manufacturer's specifications. A mention of this has been added to the text of the manuscript (line No 127). 

11) We hope for the understanding of the distinguished reviewer. The experiment was carried out at a facility built about 20 years ago. We only had access to design data on the characteristics of the materials preserved in the organization's operations department. None of the manufacturers of used materials had continued their activities by this time. To provide data of similar materials in the article would mean a decrease in the reliability of the research performed. And the study of the characteristics of materials withdrawn from an already constructed facility was almost impossible. Therefore, we limited ourselves only to the available data.

12) The volume of the drilled channels was less than 0.01% of the volume of the wall covered by the experiment. Therefore, filling the channels with any material was deemed unnecessary. Only channel entrances were isolated. A mention of this has been added to the text of the manuscript (line No. 177)

13) Indeed, it is not entirely clear from the text that the experimental design mentioned in 24 is briefly described further. The text of the manuscript has been revised so that it does not raise doubts (lines No. 146-148). 

14) The numbering of sensors in Figures 3-a and 3-b (in numbers in circles) is correct. Sensor No. 3 is located in the channel at the border of the layers of aerated concrete and insulation. A small work card with No. 3 at the bottom of Figure 3, which is fixed precisely on the cable leading to the sensor channel No. 3, is misleading. To eliminate this false impression, the unnecessary card in the figure has been removed (lines No. 191-193).

15) Unfortunately, the presentation of the essence of the calculation method practically defies a brief presentation, since it includes multiple calculation formulas. This would take 5-6 pages of text. To avoid inconvenience, the title of  reference No. 24 is given additionally in English: “The Effectiveness of Energy Saving Investments” (new reference No. 29, line No. 332). Similarly, added the English name of reference 23 (new position no. 27, line no. 328-329).

16) This is our fundamental mistake, thank you. The title of the Figure 4 has been changed from “Designed mode” toCalculated graph”.

17) This is a very difficult point in our analysis, which we only briefly touched on in the introduction. The works known to us that critically evaluate the HSCC method do not mention such an important characteristic of porous materials as channel porosity. Only the analysis of the results (complaints about moisture penetration deep into the material) indirectly indicated the existence of noticeable channel porosity. Basically, these are not even publications, but the critical statements of experts known to us in private conversations and at scientific conferences. Therefore, we considered it unreliable to cite the references in which there is no analysis of the properties of materials and a mention of channel porosity. Summing up the answer to your remark, we can say that such references “are not available”.

18) Sorry, but we did not understand the last question…

All positive research results have been achieved and proven precisely thanks to the HSCC method. Non-stationary indoor temperature control means reducing the load on the heating system and reducing the temperature only when people are away. Due to the brevity of this period, no noticeable changes in the temperature distribution occur in the depths of the building structures. This, in turn, prevents moisture migration into structures. This achieves energy savings of up to 35% (during weekends) without any adverse or undesirable consequences. This is precisely the proven goal of our research and the main content of the entire publication. We really hope that these explanations will dispel your doubts.

Round 2

Reviewer 1 Report

The manuscript has been revised well.

Author Response

Dear Reviewer. Thank you for your kind and professional assessment of the article. We are very pleased that the article received the final support of a real expert in our field of research. Your support will help our team of authors in our future work!

Reviewer 4 Report

The authors improved significantly the manuscript. Just few details need extra attention

1)Please note that figure 2 is in the middle of table 1. 

2)Figure 3b could be rotated 90 degres so it will be more intuitive to compare the position of the sensors with figure 3a.  Make it clear from where the wall was drilled (it seems it was from the interior?). In figure 3a, sensors 3 and 7 are in the opposite sides of the wall; however in figure 3b sensors 3 and 7 are close to each other. 

3) Figure 5) the title of x axis (time) is too cluttered (put every hour and make the font size bigger). To be consistent with figure 4, add units to the time axis title. 

4) Figure 6) there is a typo in "Sensor posytion" - should be position.

Author Response

Dear Reviewer. We thank you for your further kind analysis of our article. Your comments and suggestions will be useful to us not only when working on this article, but also in our subsequent publication work. Our answers are given below in accordance with the numbering of comments and suggestions in your 2nd review.

1. Sorry, but this remark from the distinguished reviewer is incomprehensible. The WORD and PDF versions of the resubmitted manuscript do not contain any semantic or formal violations in Table 1. Perhaps you have some problem with displaying the text of the manuscript on the screen.

2. a) Figures 3-a and 3-b are not very comparable geometrically, but the height in both figures is directed vertically. Therefore, rotation of figure 3b by 90º would distort the geometric connection between the figures, even if only conditional.

b) The following mention of drilling holes from the inside of the room has been added to the text: “All channels were drilled from inside the room, and sensors 1 and 6 were passed through the window frame” (line No. 176 according to the numbering in the file “materials-1161642-peer-review.pdf”).

c) As noted above, it is difficult to compare geometrically drawings 3a and 3b. Sensor 7 is glued to the inner wall surface. The sensor 3 is not located as close to it as it looks, since it is located at the end of the channel on the opposite wall surface.

3. a) We fully agree that the abundance of data in Fig. 5 clutters his field somewhat. However, for greater reliability of the data, it was decided to use the original printout of the data obtained using the measuring complex.

b) In accordance with your comment, a unit (hour) has been added to the title of the time axis of the figure (line No. 213).

4. Typo corrected, thanks (line No. 213).
